# Bridging Large Gaps in Neural Network Representations with Model Stitching

**Neil Traft**
Vermont Complex Systems Institute
University of Vermont
Burlington, VT, USA
`ntraft@uvm.edu`

**Nick Cheney**
Department of Computer Science
University of Vermont
Burlington, VT, USA
`ncheney@uvm.edu`

## Abstract

Model stitching is a technique for assembling new neural networks from the parts of existing networks, without having to re-train or fine-tune the existing weights. It has shown promise for new forms of neural architecture search, decentralized training, and transfer learning. But what are the upper bounds on this technique? Little investigation has gone into determining exactly what types of blocks can (or cannot) be stitched together, and how. In this work, we investigate the feasibility of adapting very low layers to very high layers, and stitching across different architectures, in the context of image classification models. We develop some modifications to the original stitching methods to make it possible to achieve good performance while stitching such disparate layers: (1) We interpolate the spatial dimensions of the input; (2) we propose adapters with more complex, nonlinear transformations; and (3) we propose the use of bottleneck adapters for computational efficiency. With these modifications, we are able to stitch, for example, the lower layers of a ResNet-50 to the upper layers of a Swin-Tiny, achieving ImageNet test accuracy near to the original models.[1]

## 1 Introduction

Model stitching was originally proposed as an interpretability method, for determining whether the internal representations of two different neural networks were functionally similar [11, 1]. Thus, the original stitching papers only stitch identical layers from identical architectures, and only learn linear transformations.

Since then, many works have proposed to use stitching for the practical purpose of constructing new models, not just introspecting existing models. Collectively, these works have demonstrated the promise of stitching for a diverse range of purposes (see Section 4.2 for an overview).

In particular, some of these works describe stitching diverse model architectures—e.g. stitching a CNN to a Transformer—something which is far outside the original stitching methods [27, 17, 5, 14]. They describe assembling models from blocks pulled from a diverse model zoo. But they do not describe in detail how effective it is to stitch such diverse block types, nor any limitations on the process.

In order to judge these promising applications, we seek to know exactly what is possible with model stitching, and how to make it better. Investigating which stitching methods perform better also poses useful new questions for the understanding of neural representations: Which gaps can be

---

[1]Our source code is available at `https://github.com/uvm-neurobotics-lab/stitching/tree/unireps-2025`.

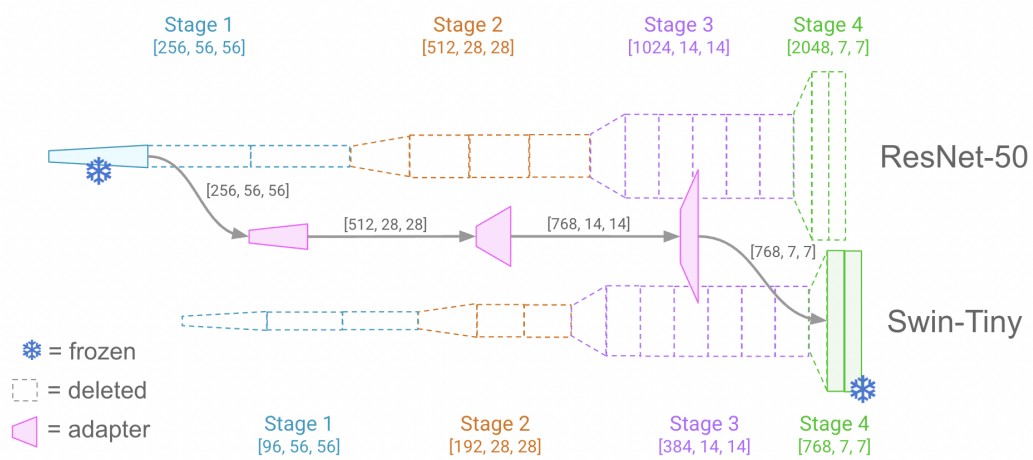

Figure 1: An example of stitching low to high layers: stitching ResNet-50 Stage 1 to Swin-Tiny Stage 4. We keep only the first and last blocks from the original models, freeze their weights, and connect them with three trainable adapters. This requires adapting tensors of size $[256, 56, 56]$ to size $[768, 7, 7]$. The final ImageNet test performance serves as a measure of how well we are able to stitch across a large gap in representation. (In this case, only a 12% drop in test accuracy, despite incurring a 65% reduction in FLOPs.)

stitched using only a linear transformation, and which ones require nonlinearities? Why do certain architectures seem more or less compatible?

Here we focus on two sub-questions: (1) is it possible to stitch at very different depths (e.g. low layers to high layers), and (2) is it possible to stitch across very different architectures (e.g. Transformer to CNN)? We find that it is, in fact, *not* possible with the original stitching methods. But with some modifications, it becomes possible and can achieve surprisingly good performance. Specifically, we find that:

1. It is possible to stitch across large gaps and across very different architectures, by interpolating the feature maps accordingly. (Section 3.1)
2. It is sufficient to use a simple bilinear image interpolation to match feature map sizes. (Section 3.1)
3. More complex adapters are needed to achieve good accuracy, rather than the traditional linear adapters. (Section 3.1)
4. Using a "bottleneck" style adapter is important for computational efficiency. (Section 3.1)
5. The BERT-style ViT architecture is less compatible to stitching than other vision transformer architectures. (Section 3.2)

With these modifications, we are able to create smaller models with competitive performance. For instance, we can compress a ResNet-50 to the equivalent of a ResNet-18 or ResNet-34 (see Table 1). Or we can compress a ViT by 33% with only a 6.4% reduction in accuracy. Or we can produce a ResNet-Swin hybrid which is 33% smaller than the original Swin, with only a 3.3% drop in accuracy.

This clarifies how well stitching can work in different scenarios, and increases the scope of what can be stitched, potentially opening up new use cases.

## 2 Methods

To shed light on how to stitch diverse architectures, we will go back to basics: **we avoid any transfer learning, and focus only on stitching two models which were trained on the same dataset.** In some cases, these might be two separate parts of the exact same model (the same set of pretrained weights); in other cases, they might have entirely different architectures. But they all share the same domain: all models were pretrained on ImageNet-1k, and all stitched models will be trained and

evaluated on ImageNet-1k. Given a wide variety of layers from the same domain, our task is to discover just how well they can be aligned.

We are given a depth $i$ within source model $A$, and a depth $j$ within destination model $B$. We denote all computations up to depth $i$ as $A_{\leq i}$, and all computations after position $j$ as $B_{>j}$. We wish to know if we can stitch these computation graphs together into a single, well-functioning graph; that is, we wish to learn a transformation $s$ such that $s(A_{\leq i}(x)) \approx B_{\leq j}(x)$. This would allow us to form a complete model $y = B_{>j}(s(A_{\leq i}(x))$.

We are particularly interested in scenarios where $i \ll j$. If such a model can achieve a test accuracy close to either of the original models $A$ or $B$, this would be a success, since it has effectively "skipped" many intermediate layers. This would show that it is possible to stitch layers at very different depths. Interestingly, this was already shown for the scenario where $i > j$ [7] (higher layers to lower layers within the ResNet family of architectures; see Section 4.1 for a discussion). But we wish to stitch lower layers to higher layers, and we intend to explore more disparate architecture combinations.

**Stages**  To reduce the possible permutations of this question, we will divide our candidate models into four stages. Many common vision models are already structured as four stages, or can be divided this way. Taking the models of Figure 1 for example: ResNet-50 consists of four stages with {3, 4, 6, 3} blocks in each stage, and Swin-Tiny consists of four stages with {2, 2, 6, 2} blocks in each stage. Typically, each stage operates at a different receptive field scale. At the beginning of each stage, the model might transform the input from $[B, C, H, W]$ to $[B, 2C, H/2, W/2]$—doubling the number of feature maps ($C$) while downsampling the size of each feature map ($H, W$)—thus, **stitching across different stages involves stitching across different spatial scales**.

We will simplify by choosing only one stitching point toward the beginning of each stage: directly after the first block in each stage, i.e., directly after the downsampling operation. We will test all pairs $x, y$ where Stage $x \leq$ Stage $y$. When $x = y$, we use different points within the same stage: $A_i$ will be after the first block, while $B_j$ will be at the end of the stage, directly before the next downsample block. See Figure 2 for a diagram of the different gaps that we will stitch across.

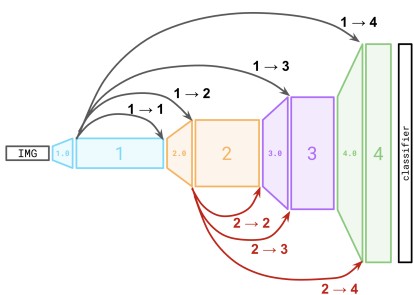

Figure 2: An illustration of the different gaps that may be stitched under our experimental protocol. For each arrow pictured here, we knock out the intermediate layers spanned by the arrow, and replace them with a particular type of adapter (or multiple adapters, in the case of bridging multiple scale-shifts).

**Architecture Combinations**  We test our ability to stitch across four significantly different architectures:

- ResNet-50: 80.4% test accuracy on ImageNet-1k [2]
- MobileNetV3-Large: 75.8% test accuracy on ImageNet-1k [3]
- Vision Transformer (ViT-Small): 78.8% test accuracy on ImageNet-1k [4]
- Swin Transformer (Swin-Tiny): 81.5% test accuracy on ImageNet-1k [5]

We attempt to stitch each architecture with itself, and also attempt to stitch across the following pairings of different architectures (a diverse, but not exhaustive, set of combinations):

- ResNet-50 $\leftrightarrow$ MobileNet
- ResNet-50 $\leftrightarrow$ ViT
- ResNet-50 $\leftrightarrow$ Swin
- ViT $\leftrightarrow$ Swin

This set of combinations covers three major vision architecture families: CNNs, BERT-style transformers (ViT), and hierarchical transformers (Swin). Each of these has a different structure to their

---

[2]Pretrained weights resnet50.a1_in1k from the Huggingface timm library [24].

[3]Pretrained weights mobilenetv3_large_100.ra_in1k from the Huggingface timm library [24].

[4]Pretrained weights vit_small_patch16_224.augreg_in1k from the Huggingface timm library [24].

[5]Pretrained weights swin_t from the Torchvision library [20].

tensors. Our code is told what structure is expected by each layer in the model, and it transforms between any pair of formats at runtime, as well as performing resizing (see below).

**Adapter Options**    The stitching methods originally used in [1] were kept as strictly linear transforms for interpretability purposes (i.e., a 1x1 convolution where each output feature map is a linear combination of the input feature maps). In our case, we are interested in finding out whatever stitching methods may be successful, even if nonlinear. Thus, we test a number of different adapters of varying complexity:

- Linear
- Linear + ReLU
- Conv3x3 + ReLU
- ResNet BottleneckBlock [6]
- Vision Transformer Block [3] (only used when stitching ViT $\leftrightarrow$ ViT)

**If we are bridging a gap across multiple stages, then we chain multiple adapters together, and train them jointly.** We use a single adapter to stitch Stage $1 \rightarrow 2$, but we use three adapters to stitch Stage $1 \rightarrow 4$. We place BatchNorm [9] layers before and after each adapter to aid in optimization, as in [1].

In addition, we also test a scenario with no adapter (fine-tuning all weights). This baseline preserves the original downsample blocks from the top model, but deletes all intermediate blocks, and fine-tunes end-to-end. Schemes similar to this have been shown to be an effective form of warm-starting [26]. However, this is only possible when stitching models with similar architecture, and cannot serve as a general replacement for stitching.

**Auto-Resizing**    We expect that we must rescale the feature maps to match the size expected at their destination. This is not *strictly* required—convolutional layers will admit varying image sizes, and transformer blocks will admit varying input sequence lengths. However, deeper layers typically expect a larger number of channels. If we grow the number of channels without shrinking the spatial dimensions, then memory usage can become unreasonably large, and often too large to fit on many common accelerators. (As shown in Section 3.1, this causes some missing results due to out-of-memory errors.)

This fact is not mentioned in "model zoo" approaches such as [27, 17, 14]. To our knowledge, StitchNet [19] is unique in mentioning this issue, but only briefly. We compare a few different methods for dealing with the spatial scaling problem:

- No Downsample — The method of prior literature.
- Downsample — Downsample just before each adapter using bilinear image interpolation.
- Integrated Downsample — Using a 3x3 convolution of stride 2.

**Optimization**    All scenarios are optimized on ImageNet-1k for just 10 epochs, using standard cross-entropy loss. We apply data augmentation to the training set: `RandomHorizontalFlip(p=0.5)` followed by `AutoAugment(policy=AutoAugmentPolicy.IMAGENET)` [2]. We use AdamW [15] with an effective batch size of 512, learning rate 0.002, weight decay 0.05, and a learning rate schedule of cosine annealing to 0.0.

We found that the fine-tuning baseline is more sensitive to learning rate, so in that case we ran with three separate learning rates (0.01, 0.002, 0.0002), and take the best result for each data point. Experiments justifying the learning rates are described in Appendix A.

For each setting, our results are aggregated over three replicates with different random seeds (the error regions in our plots are standard error over these three replicates). For some of our plots, we compute inference-time FLOPs for a single forward pass of the model. This is done with the TorchTNT library.

## 3   Results

Our results reveal that it is possible to stitch across large gaps and very different architectures—for example, achieving upwards of ~69% test accuracy on ImageNet-1k while stitching from Stage 1 to Stage 4. However, to achieve the best results we must extend the original stitching methods by (1) interpolating the feature maps appropriately, and (2) using more complex adapters with a "bottleneck" shape, rather than the standard linear adapter. We also find that the ViT architecture does not stitch as

well, especially when stitching ViT to other architectures, and propose some hypotheses for why this may be so. Note that in the following sections, standard error regions are shown on all plots, but are usually too narrow to be visible.

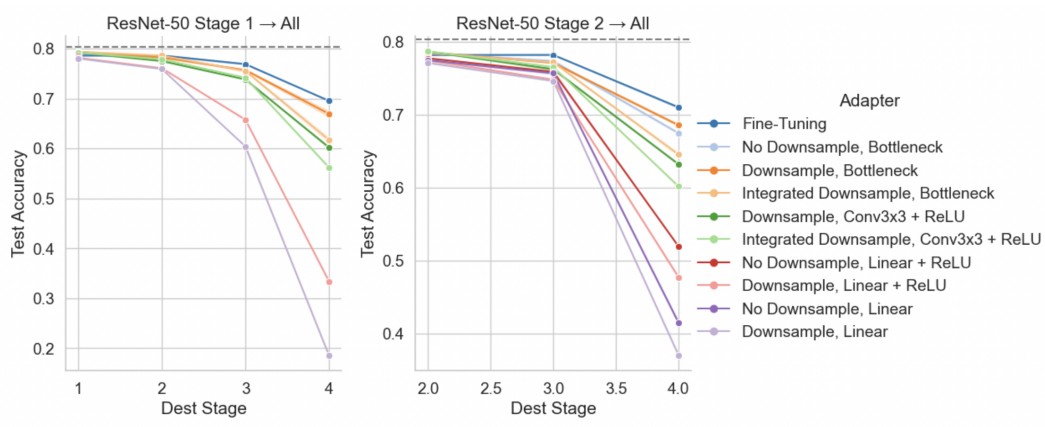

Figure 3: The results of various stitching methods on ResNet-50, across different stitching gaps. Error regions are standard error across three random seeds, and the grey line is the performance of the original ResNet-50 model. **Left:** Stitching Stage 1 to other stages. **Right:** Stitching Stage 2 to other stages. We can successfully stitch lower layers to higher layers. The smallest gap (Stage $1 \rightarrow 1$) skips 6 convolutional layers (not counting norms and ReLUs), yet only loses 1% test accuracy relative to the original model. The largest gap (Stage $1 \rightarrow 4$) skips a substantial *39 layers* (replacing 13 blocks with just 3 adapters), while still retaining >67% accuracy. Our best stitching method, *Downsample, Bottleneck*, performs comparably to the *Fine-Tuning* baseline, while having the advantage of applying beyond the use case of stitching a network to itself. Results for all other architectures and stages can be found in Appendix B.

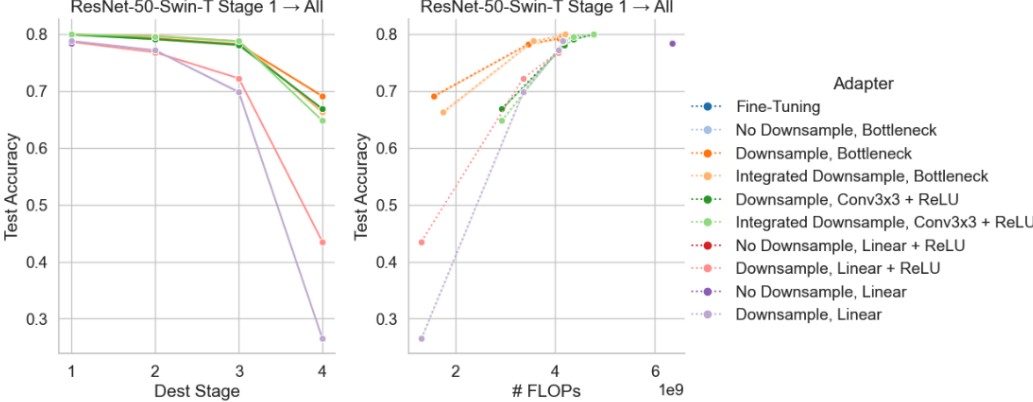

Figure 4: Stitching ResNet-50 Stage 1 into Stages {1, 2, 3, 4} of Swin-Tiny, showcasing our success at stitching different architectures (a CNN to a Transformer). Error regions are standard error across three random seeds. **Left:** Accuracy vs. stitched gap size. **Right:** Accuracy vs. inference-time FLOPs of the resulting stitched model. The original ResNet-50 and Swin-T achieve 80.4% and 81.5% test accuracy. Our best stitched models maintain >78% accuracy all the way out to Stage 3. Stage $1 \rightarrow 4$ sees a larger drop in accuracy, but uses only ~35% of the FLOPs as Swin-Tiny.

## 3.1 Stitching Across Large Gaps

We can stitch across very large gaps (e.g. Stage 1 $\rightarrow$ Stage 4), approaching the performance of full-parameter fine-tuning. (Recall that fine-tuning is only possible in the special case of stitching the same model to itself.) Without spatial scaling, stitching across such gaps is often not possible (note that the *No Downsample* variants are missing from Figures 3 and 4, due to out-of-memory errors on

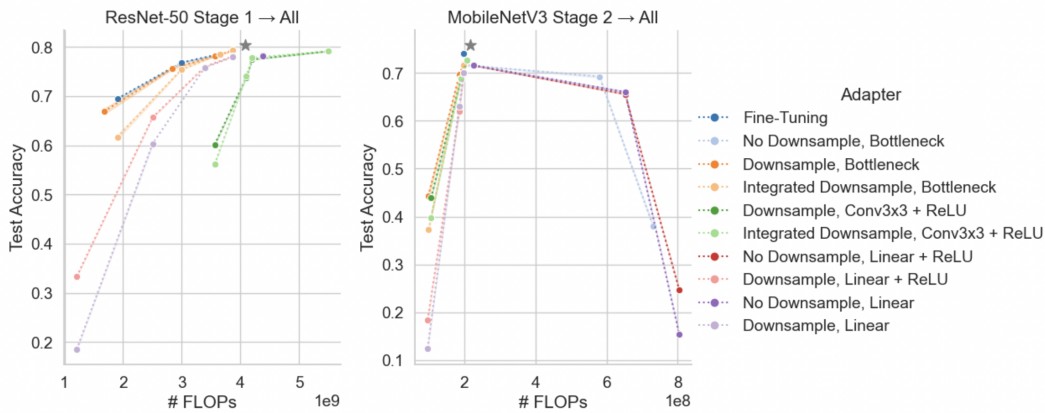

Figure 5: Accuracy vs. FLOP count, in two different scenarios. Up and to the left means better accuracy with less compute. Error regions are standard error across three random seeds, and the grey star is the performance of the original model without stitching. **Left:** ResNet-50 Stage 1. See how *Downsample, Bottleneck* matches the pareto curve of *Fine-Tuning*. Notice how the orange lines fully dominate the green lines (*Bottleneck* blocks are more efficient and performant than a full *Conv3x3*). **Right:** MobileNetV3 Stage 2. See how all *No Downsample* variants actually *grow* the computation relative to the original model, even though there are fewer layers.

an NVIDIA V100 with 32 GB VRAM when attempting to push such large feature maps into later stages of the model).

As a baseline, we can also compare to smaller ResNet variants when trained from scratch. Table 1 shows that by stitching various gaps, we can turn a ResNet-50 into the rough equivalent of a ResNet-18 or ResNet-34, with just 10 epochs of adapter training. Although the objective of this method is not to be competitive on model compression, we find this to be an intriguing example of the efficacy of stitching.

Our method functions in cases where fine-tuning cannot, and can even stitch together such different architectures as a CNN and a Swin Transformer into a very capable resulting model. Figure 4 shows that our best method (*Downsample, Bottleneck*) can stitch the ResNet Stage 1 layers

Table 1: By stitching various parts of a ResNet-50, we can produce models which are similar to the smaller ResNet variants, with only 10 epochs of training.

| Model | Accuracy | FLOPs |
|---|---|---|
| ResNet-18 | **71.5** | 1.81e09 |
| ResNet-50 Stage 1 $\rightarrow$ 4 | 67.4 | **1.67e09** |
| ResNet-34 | 76.4 | 3.66e09 |
| ResNet-50 Stage 1 $\rightarrow$ 2 | **78.3** | **3.58e09** |
| ResNet-50 | 80.4 | 4.09e09 |

into Swin Stage 3 (a ~33% reduction in FLOPs relative to Swin-T) with only a 2.2% drop in accuracy relative to the original ResNet, and a 3.3% drop relative to the original Swin model. By stitching Stage 1 $\rightarrow$ Stage 4, we can even produce a ~65% reduction in FLOPs, with only ~12% lower accuracy than Swin-T.

**More Complex Adapters**  Using more sophisticated adapters greatly improves performance over linear adapters. In Figures 3 and 4, see that *Conv3x3* and *Bottleneck* adapters substantially outperform the original *Linear* adapters used in almost all prior stitching work (all papers that we have reviewed have used either *Linear* or *Linear + ReLU* adapters).

For some smaller gaps, these linear adapters are sufficient to be competitive with more complex alternatives (for instance, see the leftmost data points in Figure 3 or Figure 4, Left). However, the performance of these adapters quickly drops off with larger gaps. These adapters are also computationally much cheaper, and this must be taken into account. But even in our Accuracy vs. FLOPs charts (Figures 4 and 5), we can see that these simple adapters are still generally dominated by the more complex adapters, in terms of their performance-computation trade-off. These trends are widely replicated across all gaps and architecture combinations, shown in Appendix B.

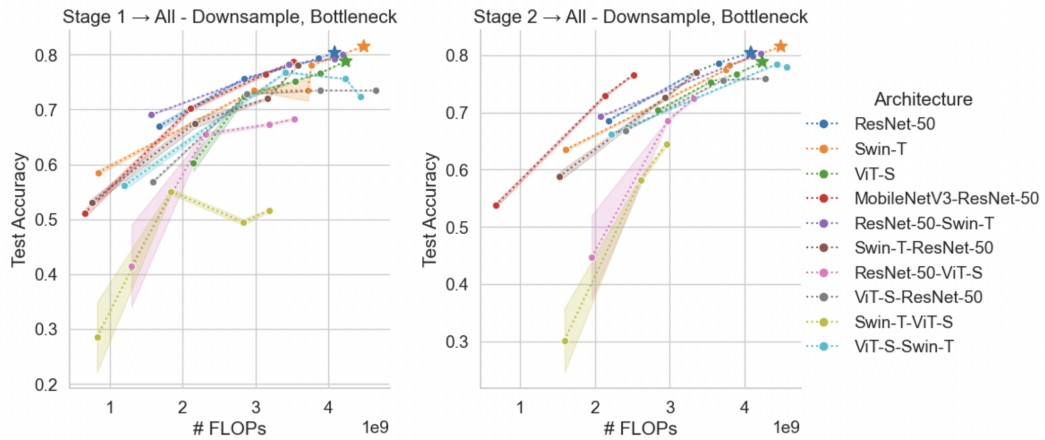

Figure 6: Accuracy vs. FLOP count, when stitching using full blocks as adapters (Vision Transformer Block for ViT-S ↔ ViT-S [the green line], BottleneckBlock for all others). This shows that many architecture combinations are approximately equally successful at stitching to each other. However, combinations involving the ViT architecture are noticeably worse, especially when ViT comes *after* another architecture. Combinations involving the MobileNet architecture are computationally cheaper. (**Left:** Stitching Stage 1 to other stages. **Right:** Stitching Stage 2 to other stages. Up and to the left means better accuracy with less compute. Stars denote the original accuracy/FLOPs of each source architecture. Certain MobileNetV3 combinations are omitted because they are on a significantly different FLOP scale. See Appendix B Figure 15 for results with these included.)

**Bottleneck-shaped Adapters for Efficiency**    Thus, increased complexity must be balanced with computational efficiency. The ResNet BottleneckBlock [6] linearly projects channels down to a smaller dimensionality before performing 3x3 convolution. Using a bottleneck rather than a full-channel convolution (*Conv3x3*) matches performance with a much smaller FLOP count (Figure 3 Right, and Figure 5 Left, orange lines vs. green lines). A similar effect was used in [5], but it was based on LoRA [8] and limited to a linear transformation. Naive downsampling (as opposed to "Integrated" downsampling) is also important for keeping the model tractable. See Figure 5.

**Integrated Scaling is Not Needed**    We also considered the possibility that interpolating from one receptive field scale to another would need to be learned, and thus it should be integrated into the adapter itself. To test this, we used adapters which incorporate a strided convolution, such that scaling and transformation happen simultaneously. We name these adapters as "*Integrated Downsample*".

We find that this integrated downsampling is not necessary; a separate downsample step works just as well—in fact, better. In Figures 3 and 4, we see that *Downsample, Conv3x3* beats *Integrated Downsample, Conv3x3*, and similarly for the *Bottleneck* adapters. Separate downsamples almost always perform better, and this replicates across all 12 scenarios in Appendix B.

## 3.2   Cross-Architecture Comparison

Figure 6 shows a comparison of the best stitching results for each architecture pairing (in other words, the results of stitching with full bottleneck block adapters). We can see that many architecture combinations are approximately equally successful at stitching to each other—they generally rest on the same pareto front, within a few accuracy percentage points.

The most clear shifts away from the pareto front all involve the original Vision Transformer (ViT) architecture, especially where ViT is stitched on top of another architecture (*ResNet-50-ViT-S* and *Swin-T-ViT-S*). Recall that ViT is a BERT-style architecture. We hypothesize two factors that contribute to ViT's poor stitchability.

First, the ViT architecture uses an extra token, the class token, added to the beginning of every input sequence. This token is used as the sole output for the classifier head. When attempting to stitch an architecture without a class token to one that has one, we simply take the mean of all other tokens as the class token. When going the other way, we concatenate the class token onto every other token

(effectively doubling the number of channels). The adapter then has the task of determining how to disentangle/integrate class token information from/into all other tokens. This proves to be a challenge.

Second, the ViT architecture concatenates a position encoding onto each token at the very beginning of the forward pass. This encoding then becomes entangled with all other token information as it passes through the model, so it is not obvious how to provide this information to a hidden layer partway through the model. This differs from many other more recent Transformer models for vision (including Swin) which add a position encoding just before each attention layer [25, 13, 28, 21, 12].

The fact that Swin stitches very successfully to ResNet leads us to recommend focusing on Swin or similar hierarchical transformer architectures when stitching across vision architectures. However, this poor performance is highly dependent on our implementation choices, and it may be possible to find improved ways of transforming between BERT-style sequences and image-like formats.

## 4 Related Work

### 4.1 Flexibility of Stitching

Prior work has called into question the original application of stitching. In the process, this has shown stitching to have much more potential capability than originally imagined.

As stated in the Introduction, model stitching was originally devised as a test of functional similarity [11, 1]. The idea is that, if a linear mapping from one layer to another can be learned such that the behavior of the original network is recovered, then those two layers must have the same representation, up to an affine transformation.

However, subsequent works have called this idea into question. Hernandez et al. [7] perform some limited experiments similar to ours. They show that lower layers can be successfully stitched onto higher layers ($i > j$ in our notation). This suggested that stitching could be successful even when the representations do not encode "similar information". This shows promise for the flexibility of stitching and motivates our work, which examines the stitchability of lower to higher layers ($i < j$), as well as the stitchability across more disparate architectures.

Similarly, Smith et al. [18] show an array of cases where stitching is unexpectedly successful. They even suggest the possibility of switching input modalities via stitching: they stitch an ImageNet-trained model into a model trained on a 10-class dataset of bird songs. This begs the question, "What *can't* be stitched?", which we explore in our work.

### 4.2 Potential Applications of Stitching

Already, many works have demonstrated the promise of stitching for a diverse range of purposes. Deep Incubation [16] enables more distributed training of large models by training parts of the network separately and stitching them together later. FedLego [22, 23] uses stitching to enable the use of heterogeneous models in federated learning. Other works select subsets of a number of models from a model zoo, and stitch them into a new model; this suggests a more efficient, flexible form of transfer learning compared to fine-tuning a single model [27, 5, 14, 19]. Even setting aside transfer learning, borrowing such diverse modules from a model zoo can still be useful for other things such as model compression [17], architecture search, or ensembling [4]. It is these works which describe stitching an open-ended set of modules which most inspire our work.

Thus, stitching has promise for a diverse range of use cases. It may enable new forms of transfer learning, modular deep learning, or recycling of neural modules [10]. In order to judge these promising directions, we seek to know exactly what is possible with model stitching, and how to make it better.

## 5 Conclusion & Future Prospects

We have determined two key modifications which greatly increase the scope of layers that can be stitched together: (1) using more complex, but efficient, transformations; and (2) interpolating feature maps to their expected scale and structure. We achieve accuracies on ImageNet-1k comparable to those of models trained from scratch, with only 10 epochs of adapter training. This reveals stitching

as a promising technique for assembling novel architectures from reusable components, and invites the design of new adapters to further expand the scope of what can be stitched.

We have explored how different architectures can be joined together, but we are still curious how stitchable are models trained on different domains. This will be useful knowledge for applying these techniques to transfer/multi-task learning. Also, the cases for chaining modules in series may be somewhat limited. It would be even more useful to be able to join modules in parallel, to allow usage of multiple modules simultaneously. Nevermind the fact that this work begs various questions about *why* it is possible to stitch such disparate representations. These are only some of the compelling avenues for future work in model stitching.

## Acknowledgments and Disclosure of Funding

This material is based upon work supported by the National Science Foundation under Grants No. 2239691 and 2218063. Computations were performed on the Vermont Advanced Computing Core supported in part by NSF Award No. OAC-1827314.

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

# A    Learning Rate Study

We find that training stitching adapters is relatively stable with AdamW. Our chosen learning rate 0.002 and weight decay 0.05 seem to be optimal across the various adapters and architectures that we checked. See Figure 7.

However, for fine-tuning all weights, we find larger performance differences across different learning rates (Figure 8). We also find the best learning rate is not consistent across all scenarios. Due to these issues, we run separate trials for all three learning rates and report the best performer in each unique scenario (see Section 2, subsection "Optimization").

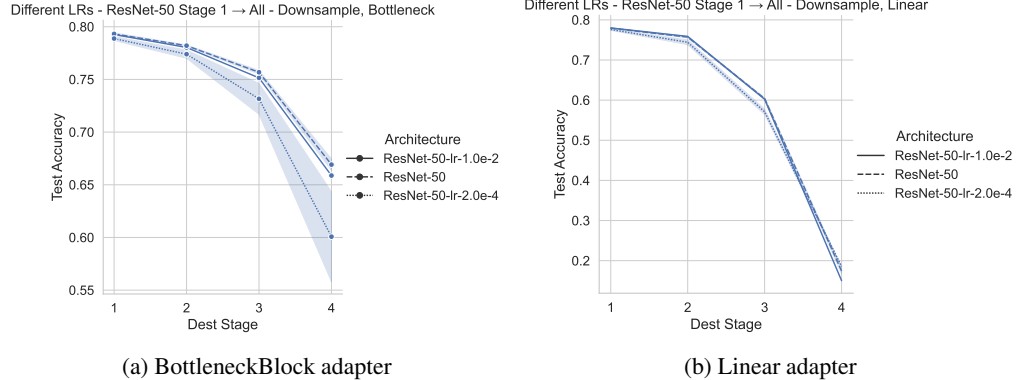

(a) BottleneckBlock adapter                              (b) Linear adapter

Figure 7: Optimizing adapters using various learning rates, to stitch ResNet-50 Stage 1 to others. We find that the default learning rate (the dashed line) is generally optimal, with minor differences to other learning rates.

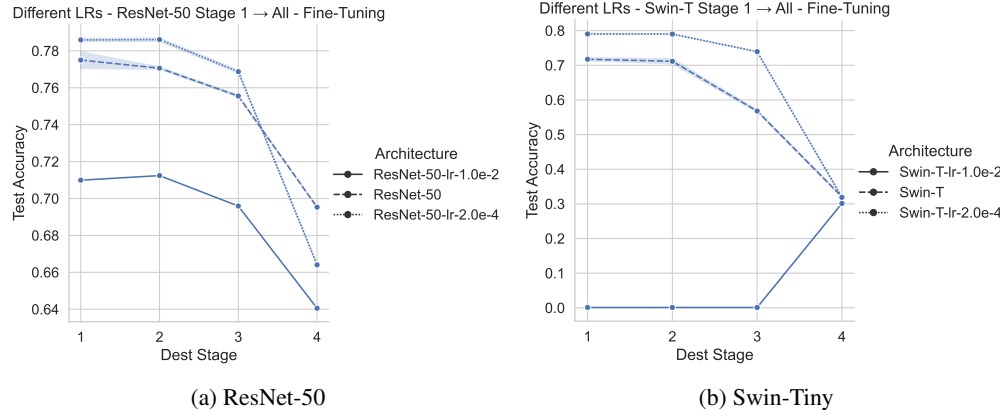

(a) ResNet-50                              (b) Swin-Tiny

Figure 8: Fine-tuning all weights using various learning rates, on two different architectures. In contrast to the adapters, we find a larger performance difference across different learning rates, especially for Swin-T. We find that fine-tuning tends to prefer the smaller learning rate (the dotted line), but the best learning rate is not consistent across all scenarios. Due to these issues, we run separate trials for all three learning rates and report the best performer in each unique scenario.

# B    Full Results

See Figures 9, 10, 11 for accuracy vs. depth results across all scenarios. See Figures 12, 13, 14 for accuracy vs. FLOPs results across all scenarios.

Figure 15 shows a comparison of the best stitching results for each architecture, including the MobileNetV3 variants which were omitted from Figure 6 for visibility.

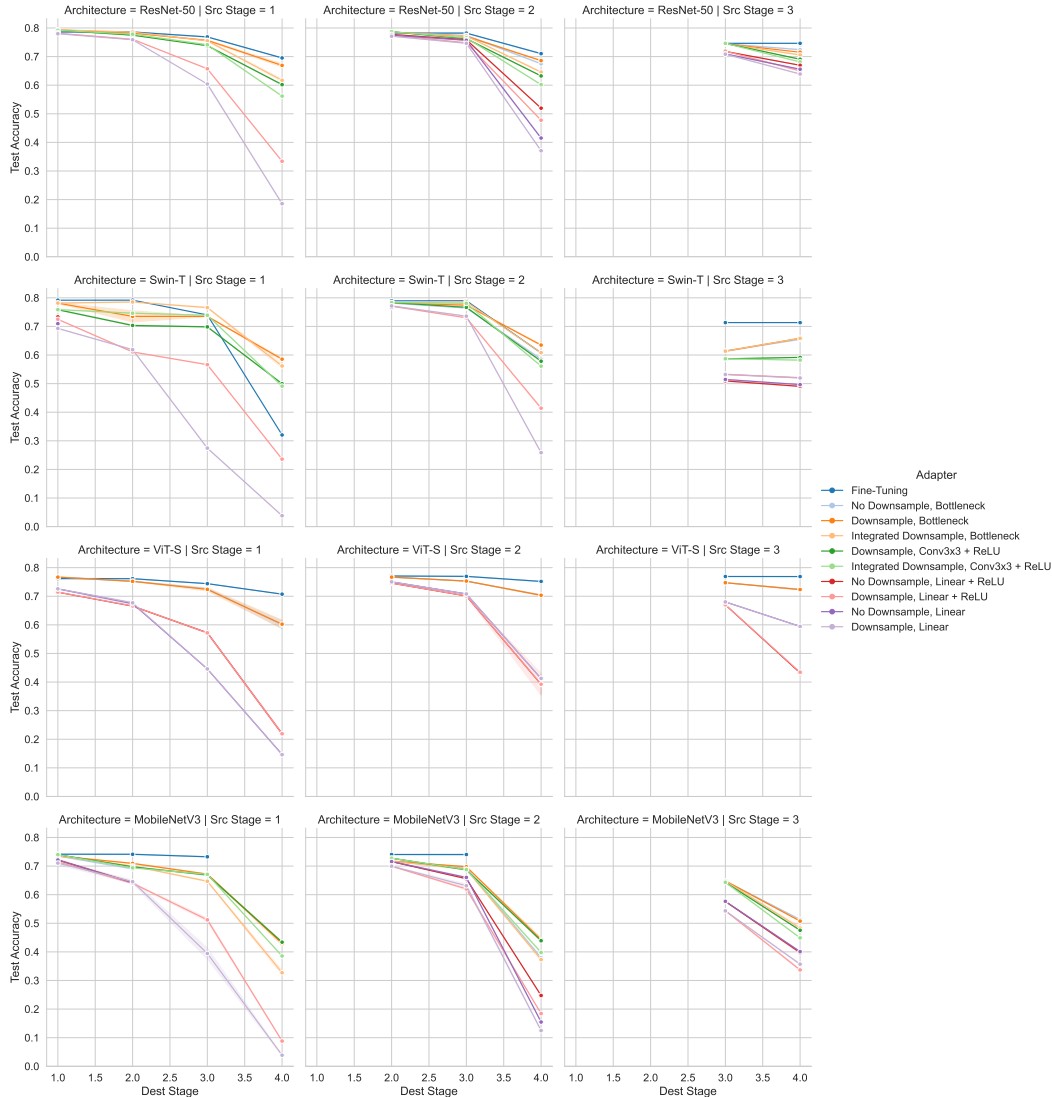

Figure 9: Accuracy vs. stitching gap size, across three stages and four architecture combinations. (Part 1) (*Note:* In the "Architecture = ViT-S" plots, the line named "Bottleneck" is actually the Vision Transformer Block, renamed to simplify the legend. This is the only place where ViT Blocks were used as adapters.)

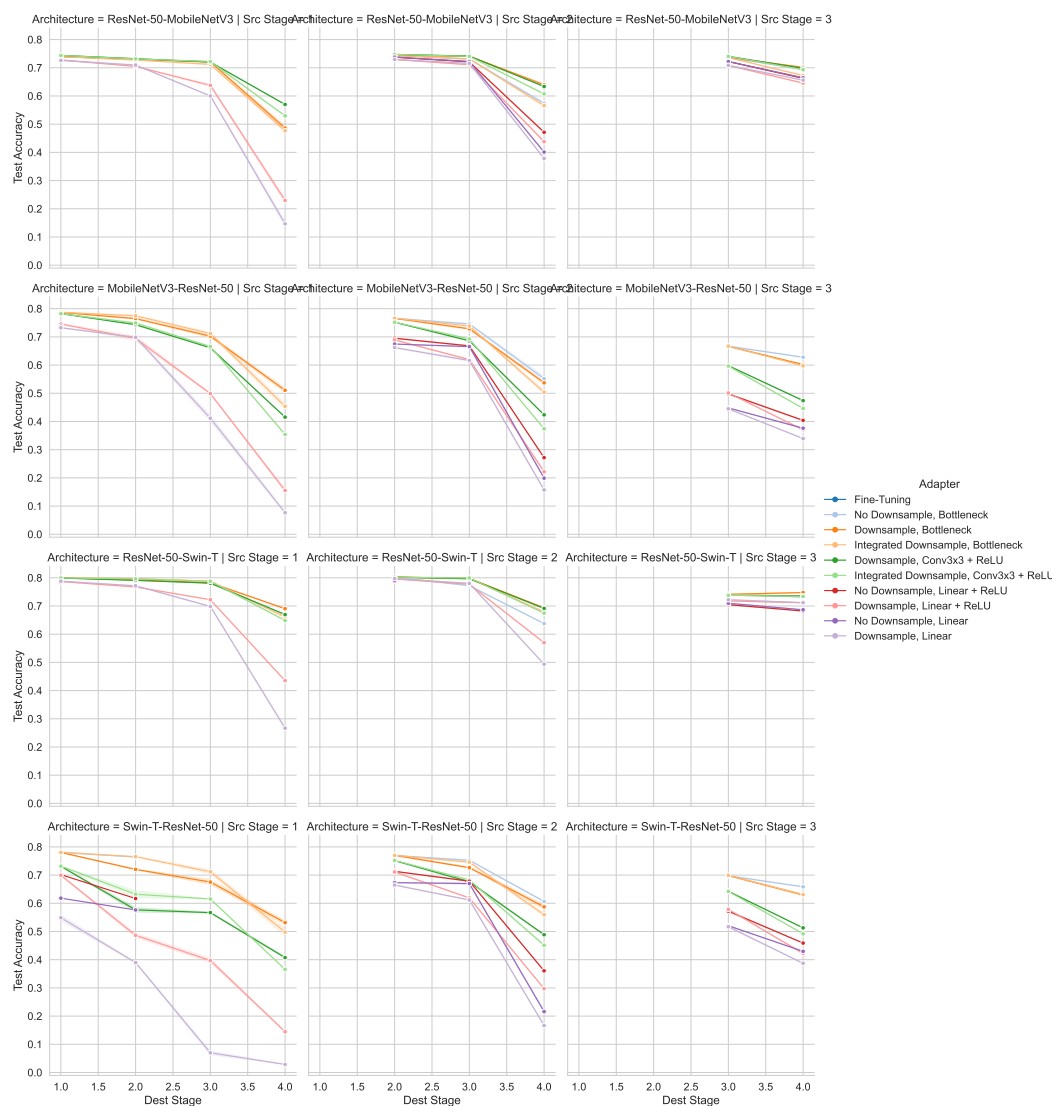

Figure 10: Accuracy vs. stitching gap size, across three stages and four architecture combinations. (Part 2)

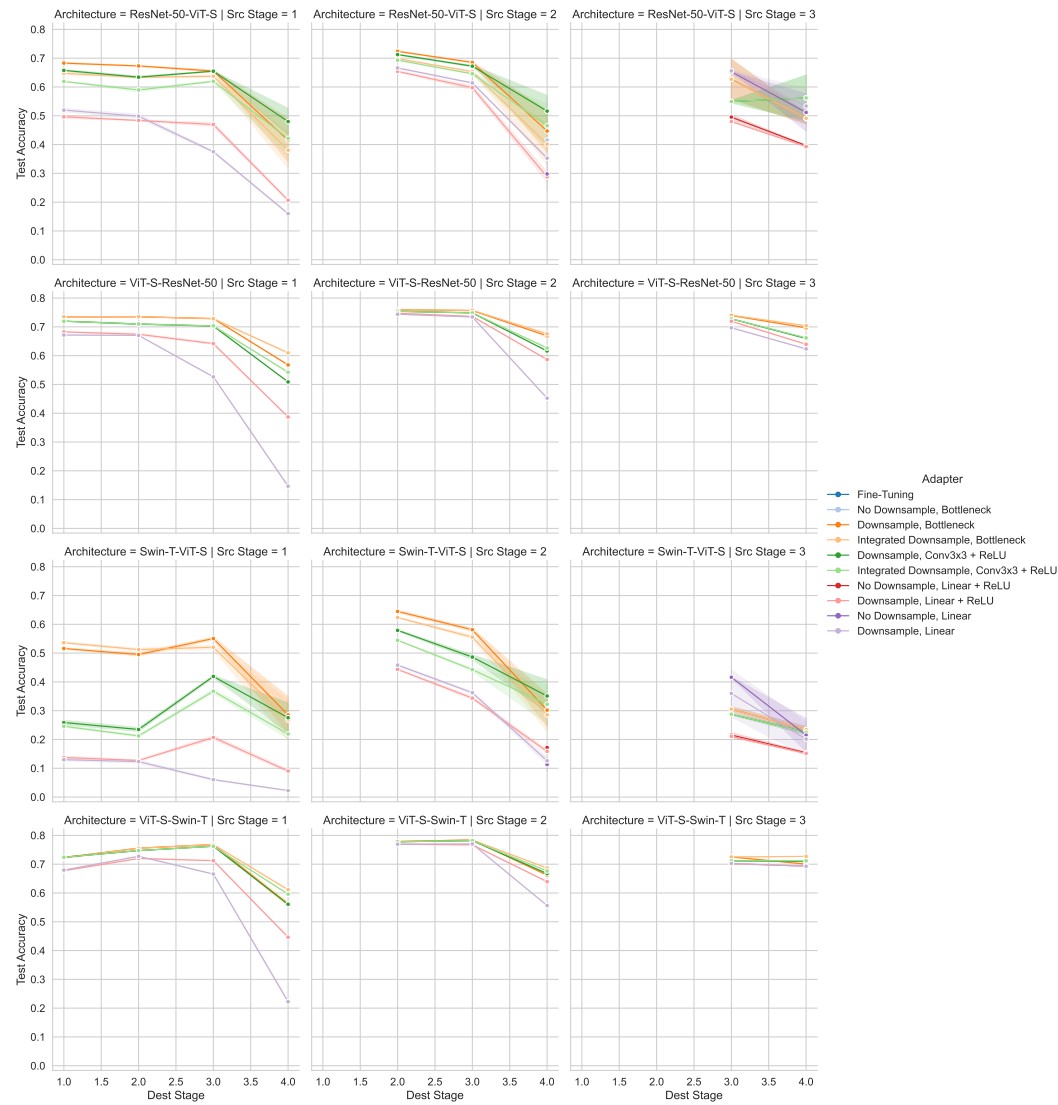

Figure 11: Accuracy vs. stitching gap size, across three stages and four architecture combinations. (Part 3)

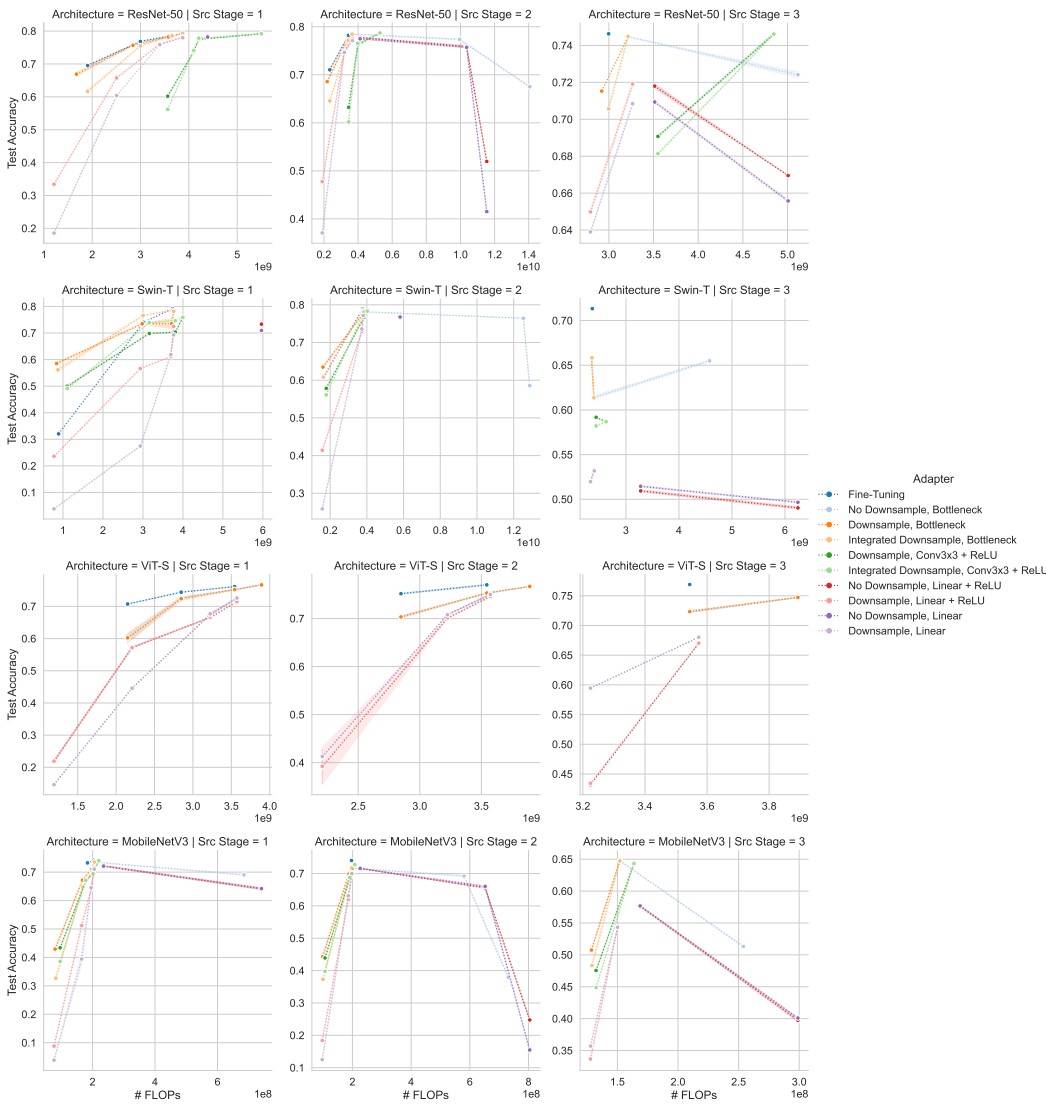

Figure 12: Accuracy vs. FLOP count of the resulting model, across three stages on all four architectures. (Part 1) (*Note:* In the "Architecture = ViT-S" plots, the line named "Bottleneck" is actually the Vision Transformer Block, renamed to simplify the legend. This is the only place where ViT Blocks were used as adapters.)

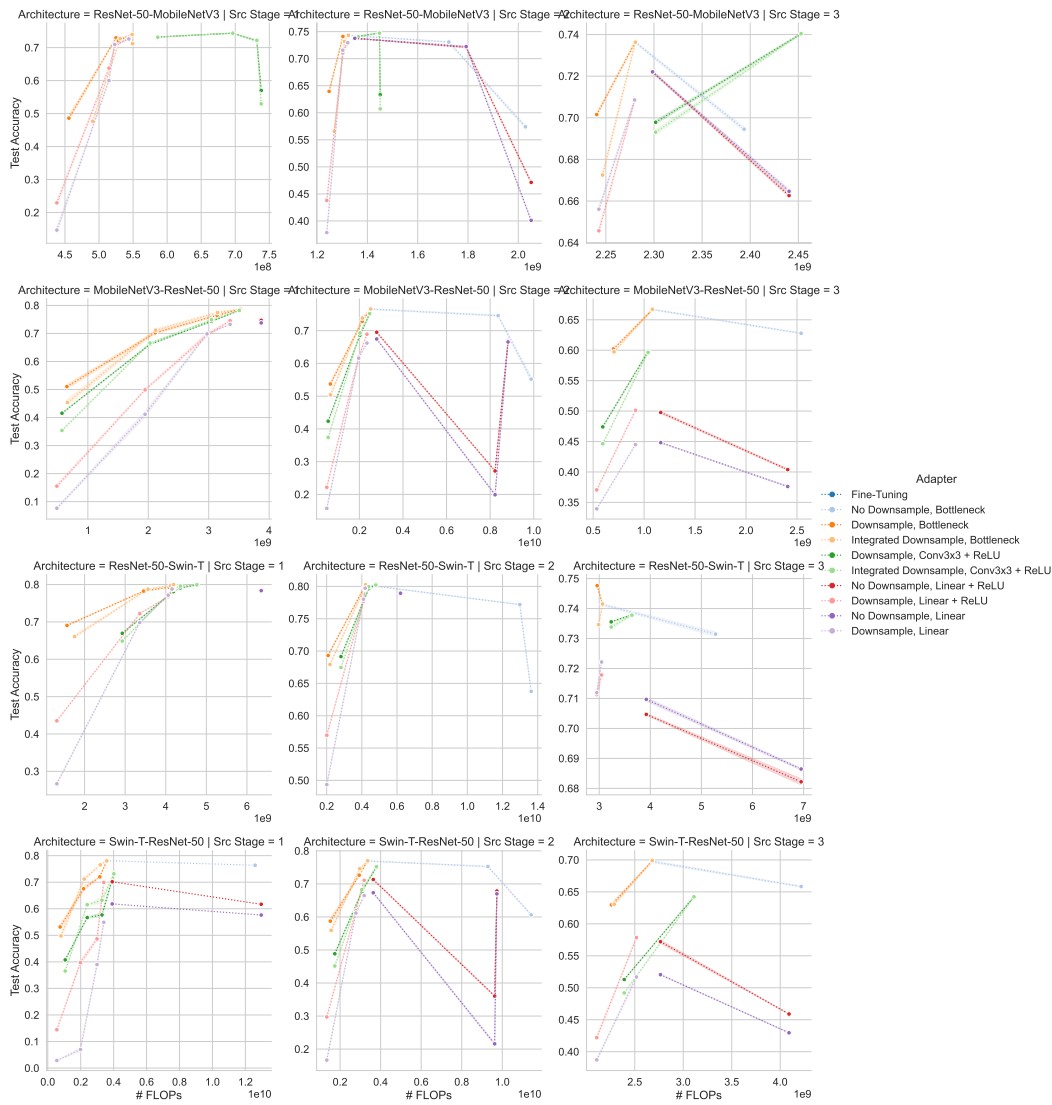

Figure 13: Accuracy vs. FLOP count of the resulting model, across three stages on all four architectures. (Part 2)

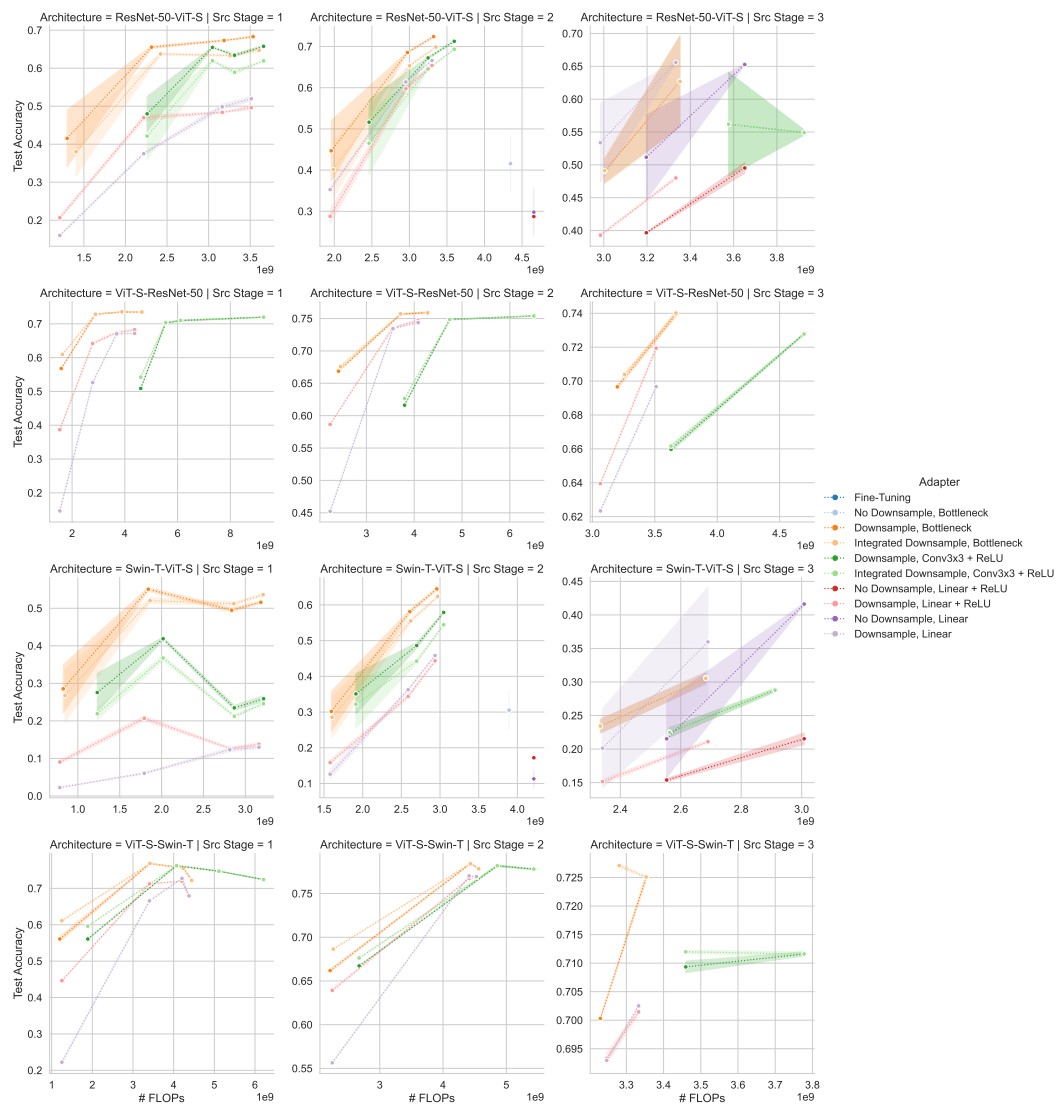

Figure 14: Accuracy vs. FLOP count of the resulting model, across three stages on all four architectures. (Part 3)

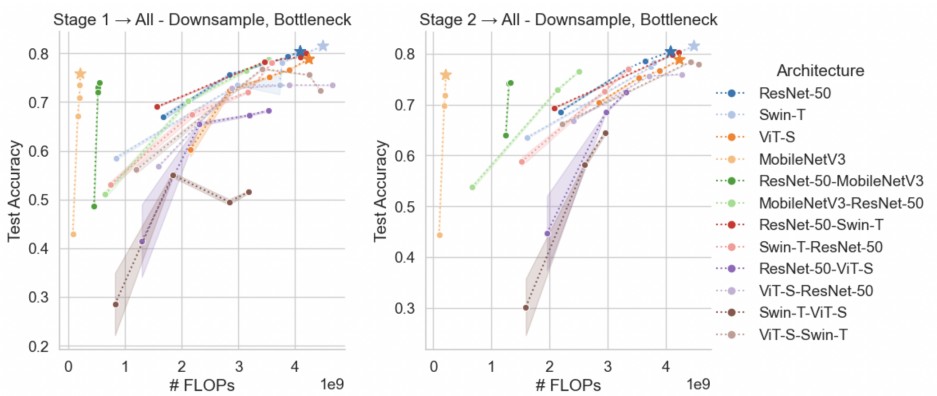

Figure 15: Accuracy vs. FLOP count, when stitching using full blocks as adapters. (Up and to the left means better accuracy with less compute. Stars denote the original accuracy/FLOPs of each source architecture. **Left:** Stitching Stage 1 to other stages. **Right:** Stitching Stage 2 to other stages.)

