# OpenReview forum: "Bridging Large Gaps in Neural Network Representations with Model Stitching"
_NeurIPS.cc/2025/Workshop/UniReps — UniReps2025_

### Official Review · Reviewer_gzbb · 2025-09-11

**Confidence:** 4

**Review:**

## Short Summary:

This submission introduces a model stitching method for combining neural network components from different model architectures or stages without end-to-end retraining. Unlike previous studies that require compatible model architectures or similar layer types, this work experiments with stitching very low to high layers and across CNN to Transformer models by incorporating several technical improvements (e.g.,  spatial interpolation of feature maps, nonlinear and bottleneck adapters). Experiments show that effective stitching is possible even from ResNet-50 to Swin-Tiny, with only moderate performance drop. Overall, this submission offers new perspectives in practice. However, there is a significant mismatch between the paper title's broad claims and the narrow experiment scope (solely on image classification tasks).

---

## Strengths:
**(S1)** This work targets a timely and practical problem on computer vision model assembly and transfer without extensive full re-training. The experiment setup is reasonable, covering diverse model families (ResNet, MobileNet, ViT, Swin Transformer) and layer depths with both within-architecture and cross-architecture validation on the widely-used ImageNet-1K.

**(S2)** It identifies specific limitations of existing stitching methods and proposes actionable improvements, such as feature map interpolation and the use of nonlinear and bottleneck adapters for computational efficiency. These designs are well motivated by empirical results, which could serve as a clear guidance for future model stitching studies. I think this represents the most significant contribution of this work.

**(S3)** Figures and tables are detailed and informative, which presents the empirical findings clearly. In particular, **Figure 3** shows the accuracy degradation and computation trade-off across stitching gaps and adapter choices, and **Figure 4** shows the potential to stitch different architectures (e.g., from a CNN to a Transformer) with strong performance under the proposed modifications. **Figure 6** and related discussion in Sec. 3.2 reveal the particular difficulty of stitching into ViTs, with sensible hypotheses about class token and position encoding incompatibilities.

**(S4)** The manuscript exhibits a strong logical flow. Unlike usual papers, this manuscript makes extensive use of sub-sections and bold paragraph headings, which is more like a technical report. Note that this is not a disadvantage but is well-suited for this work, which is with straightforward logic and does not require lengthy story telling. However, there are still several issues in the presentation of problem formulation and technical details in methods and experiments, which I have listed below.

---

## Weaknesses:

**(W1)** Mismatch between paper title and experiment scope. The title “Bridging Large Gaps in Neural Network Representations with Model Stitching” suggests broad applicability. However, all experiments in the manuscript are on computer vision tasks, specifically the ImageNet-1K classification benchmark. This cause sort of overclaiming as there are large domains and modalities in representation learning that have not been touched. It leaves open the question of whether the findings are robust to other domains or not. To address this, I strongly recommend revising the title and abstract to reflect the focus on computer vision models.

**(W2)** Within computer vision, the experiments still leaves certain baselines and analysis under-explored. For example:

- **(W2.1)** Comparison against other modular model composition approaches beyond stitching and fine-tuning is missing, such as model merging methods [1] [2] [3].
- **(W2.2)** The experiments are limited to image classification. The practical applicability of the findings would be significantly supported by showing that stitched models are effective on other downstream vision tasks (e.g., object detection, semantic segmentation), which is a key evaluation metric for general-purpose models [4] [5].
- **(W2.3)** Ablation on adapter training. In this submission, adapters are trained for only 10 epochs by default. It lacks an ablation study on the training protocol and hyper-parameter sensitivity, as well as their impact on performance, which would demonstrate the robustness of adapter designs.
- **(W2.4)** For ViT-specific failure cases, the discussion in Sec. 3.2 is mostly heuristic. There are no experiments like, IMHO, with ablated or alternative class token/positional encoding handling to support the hypotheses. While not necessary, this would be helpful and may greatly inspire future research.

**(W3)** Specific comments on Figures.

- **Figure 3, 4:** These strongly support the claim that advanced adapters and feature interpolation are critical. However, they also reveal that performance degrades sharply across the largest gaps (e.g., Stage 1→4), and that simple linear adapters are insufficient except for very small gaps. This seems to be contradict to the claim of broad stitchability.
- **Overall Figure Quality:** The figures appear to be low-resolution images with blurred texts rather than vector graphics. I recommend the authors to use high-resolution vector formats (e.g., PDF, SVG) for all figures for better readability.

**(W4)** To immediately engage the reader, I recommend giving more specific and compelling results in Sec. 1 introduction, which presents the paper's findings in a high-level manner. For example, citing a key result from the start (e.g., "our method stitches between identical ResNet-50 stages, skipping 6 convolutional layers with only a 1% drop in test accuracy") would help readers quickly grasp the core contributions of the work. Now it is somewhat too brief.

**(W5)** Typos and factual issues.

- **Line 77:** ResNet-50 actually consists of four stages with {3, 4, 6, 3} blocks, not {3, 4, 6, 4} as written in the manuscript.
- **Line 127:** I suppose the term "graphics cards" should be replaced with formal "graphics processing units (GPUs)".

**(W6)** Formatting and Layout. Several paragraphs end with very short lines containing only one or two words (like lines 27, 37, 56, 91, etc.). This creates excessive white space and disrupts the reading flow. Thus, I suggest rephrasing sentences to create a more compact layout.

**(W7)** As stated in **(S4)**, I recommend adding a subsection in Sec. 2, such as 2.1 Problem Definition to formally set up the problem. Furthermore, existing formulation is a bit abstract. It would be more accessible if it can be grounded in a practical definition—for instance, by explaining that stitching replaces a block of layers in one model with a learnable adapter that transforms features from another model. In addition, using Figure 1 as an example to illustrate this process would greatly help understanding. IMHO, now the description in lines 85-89 is difficult to follow.

**(W8)** The explanation regarding stitching lower-level layers to higher-level ones and vice versa (lines 67-70) is unclear. Does this apply only to identical architectures? When stitching between different models (e.g., a CNN and a Transformer), this would imply 4 distinct scenarios. The authors should provide a more precise definition and a clarifying example for better clarity.

**(W9)** The "with no adapter (fine-tuning all weights)" baseline (lines 120-124) is a bit confusing. Since stitching replaces weights with adapters, it is unclear what "fine-tuning all weights" refers to. Does this baseline involve combining two model segments and then fine-tuning all of their parameters jointly? Please clarify this procedure.

**(W10)** Lines 125-126 state that feature maps are rescaled to prevent "unreasonably large memory usage", but the subsequent explanation in Sec. 3 is not provided. This justification seems incorrect. To my understanding, for hierarchical networks like ResNet, feature maps between stages have incompatible feature sizes, making resizing a necessity to build a valid computational graph. Thus I fail to catch the statement of “memory usage can become unreasonably large”. Similar issues occur throughout the manuscript. I believe this submission targets a problem with both research and practical value, and its findings are meaningful. However, many technical details remain unclear.

**(W11)** Several references should be included in related work for better literature review.

- Similarity of Neural Network Representations Revisited [6] offers analysis of neural representation similarity which serves as the foundation for understanding stitchability.
- Stitching for Neuroevolution: Recombining Deep Neural Networks without Breaking Them [7] explores methods for recombining trained networks via stitching. This aligns with this submission’s aims of recombining model architectures.

---

### Reference

[1] Editing Models with Task Arithmetic, ICLR 2023

[2] TIES-Merging: Resolving Interference When Merging Models, NeurIPS 2023

[3] Task Arithmetic in the Tangent Space: Improved Editing of Pre-Trained Models, NeurIPS 2023

[4] Git Re-Basin: Merging Models modulo Permutation Symmetries, ICLR 2023

[5] Twin-Merging: Dynamic Integration of Modular Expertise in Model Merging, NeurIPS 2024

[6] Similarity of Neural Network Representations Revisited, ICML 2019

[7] Stitching for Neuroevolution: Recombining Deep Neural Networks without Breaking Them, arXiv 2024




---
## Justification & Message to ACs and Authors:

Specific questions and advice are already provided in **Strengths** and **Weaknesses**. Although there is no rebuttal stage, I still encourage the authors to refine the manuscript based on the comments to further strengthen this work, regardless of its final acceptance status.

I first give a Weak Accept (3) to **recognize the potential practical value** of this work, particularly the empirical insights for specific stitching adapter designs. However, **significant issues remain** as mentioned above, especially for the unclear technical details and writing issues. IMHO, the manuscript's clarity and technical rigor do not meet the standards of a top-tier conference paper. Thus, I believe this submission requires special attention from ACs to determine whether it meets UniReps' standards.

Overall, I hope these comments help fellow reviewers and ACs understand the basis of my recommendation. I am open to the follow-up discussions to reach a consensus for the final recommendation.

**Score:**

3

**Topic Fit:**

3

---

### Official Review · Reviewer_jfHQ · 2025-09-11
**Good paper, but raises questions about the role of the nonlinear adapter.**

**Confidence:** 4

**Review:**

This paper explores the limits of model stitching, focusing on stitching low-level to high-level layers and across disparate architectures (e.g. CNN→Transformer). The authors introduce feature map interpolation for spatial alignment and nonlinear adapters to bridge complex shifts. While the experiments are solid and the conclusions compelling, I question whether these nonlinear adapters act more as lightweight surrogates for the skipped intermediate modules, rather than truly testing whether residual features can be transformed and stitched directly.

Given this concern, I suggest the authors include a case study comparing the features captured by the learned adapter with those from the original intermediate layers. Such analysis may reveal how the adapter functions and whether the skipped module can be omitted without significant performance loss.

**Score:**

4

**Topic Fit:**

3

---

### Official Review · Reviewer_DpHw · 2025-09-13
**Clear and solid paper exploring models stitching across many layers or different architectures.**

**Confidence:** 3

**Review:**

The paper focuses on model stitching, i.e., assembling a neural network from multiple other networks. Specifically, it explores how feasible it is to adapt layers from different parts of the network or very distinct architectures. The authors show that it is possible with the use of more complicated adapters.

Strengths:
- The topic is interesting and well-motivated; the authors show a clear gap in the literature.
- The paper is well-written with a clear structure and is easy to read.
- Results show an improvement of the state-of-the-art and more prospects of using this method in the future.

Weaknesses:
- There are some things from the methodology that are not clear to me. It might perhaps be a good idea to clarify those:
	- Why are you stitching only four pairs of models, not all six variants (in other words, why are pairs MobileNet-ViT and MobileNet-Swin missing)?
	- You mention different stages (lines 72-84). How do you define these stages for the other networks you are using in your experiments, not only ResNet?
	- You mention "Vision Transformer Block" as one option for the adapter, but you do not report any results on this. Why?
	- Why is it necessary to have one adapter per stage? Why can't we use a single adapter to stitch Stage 1->4?

**Score:**

4

**Topic Fit:**

3

---

### Official Review · Reviewer_H5dc · 2025-09-16
**Cross-architecture model stitching works surprisingly well with simple resize + bottleneck adapters; strong breadth, but needs stronger baselines and fuller reporting to clear proceedings bar.**

**Confidence:** 4

**Review:**

## Summary
The paper systematically studies **model stitching** across large depth gaps and heterogeneous architectures (ResNet-50, MobileNetV3-L, Swin-T, ViT-S) on ImageNet-1k. The practical recipe is: (1) explicit **bilinear resizing** of intermediate features, (2) **nonlinear adapters** with a preference for **ResNet bottleneck** blocks, and (3) chaining adapters for big gaps. With only **10 training epochs** for adapters, many stitchings (especially ResNet→Swin) retain strong accuracy with substantial FLOP savings, while **ViT as the top network** is notably harder to stitch—hypothesized to be due to class token and positional encoding entanglement.


## Strengths
* **Well-posed empirical question** with a clean, reproducible protocol (evaluate all stage pairs, train adapters only).
* **Breadth**: same-arch and cross-arch stitchings; multiple adapter types; compute/accuracy trade-offs.
* **Actionable takeaways**: explicit bilinear resize is key; **bottleneck adapters** Pareto-dominate linear/3×3 in many cases; separate resize outperforms “integrated” stride-2 convs.
* **Candid negative evidence** for ViT-top stitchability, with plausible hypotheses.
* **Potential impact** for modular re-use and unified representations in practice.


## Weaknesses
* **Positioning vs related work** needs sharpening (model re-assembly/stitching/representation alignment): clarify novelty and provide matched-budget comparisons where possible.
* **Baselines** are thin for cross-arch cases (no **knowledge distillation** or **shallow projection + head**); same-arch misses **partial fine-tuning** of the last kept block + head.
* **Training budget**: only **10 epochs**; lack of duration sensitivity may understate some adapters or overstate gaps.
* **Metrics**: efficiency claims would benefit from **adapter parameter counts, latency/throughput, and memory** (incl. OOM cases).
* **Reproducibility**: experiments use only two seeds without reporting mean±std; details on hardware are sparse and the data preprocessing pipeline is not described.
* **Analysis depth**: ViT failure analysis is speculative; small diagnostics (class-token handling, positional encoding variants, token-to-2D spatialization) would strengthen insight.


## Questions for the Authors
* Sensitivity to **adapter width/depth** (bottleneck ratio; number of chained adapters)?
* Effect of **normalization placement/type** around adapters in cross-arch stitching?
* Does **pre-initializing stride-2 convs** to approximate bilinear close the gap to explicit resize?
* Can **token spatialization** or **alt positional encodings** mitigate ViT-top issues?
* What are the adapter parameter counts, wall-clock training times, and memory footprints?


## Evaluation
* **Originality:** Good—systematic breadth + simple, effective recipe; conceptual novelty modest.
* **Technical soundness:** Solid experimental design; needs stronger baselines, more seeds, and duration sensitivity.
* **Clarity:** Clear writing/figures; add numeric callouts/tables for key Pareto points.
* **Significance:** Provides practical guidance for modular model reuse and contributes to unified-representation research, while highlighting challenges with ViT that point to important open problems.
* **Completeness of results:** Broad, but missing efficiency metrics and robustness stats.


## Overall Recommendation
**Weak Accept.** A valuable empirical contribution with practical guidance. The paper would benefit from stronger baselines and fuller reporting/diagnostics, but the current submission already meets the bar.

**Score:**

3

**Topic Fit:**

3